# Predicting lung aging using scRNA-Seq data

Qi Song[1], Alex Singh[1], John E. McDonough[2], Taylor S. Adams[3], Robin Vos[4], Ruben De Man[3], Greg Myers[4], Laurens J. Ceulemans[5], Bart M. Vanaudenaerde[5], Wim A. Wuyts[5], Xiting Yan[3], Jonas Schupp[3], James S. Hagood[4], Naftali Kaminski[3], Ziv Bar-Joseph[1,6]*

1 Computational Biology Department, School of Computer Science, Carnegie Mellon University, Pittsburgh, Pennsylvania, United States of America, 2 Faculty of Health Sciences, McMaster University, Ontario, Canada, 3 Section of Pulmonary, Critical Care, and Sleep Medicine, Yale University School of Medicine, New Haven, Connecticut, United States of America, 4 Department of Pediatrics (Division of Pulmonology) and Marsico Lung Institute, University of North Carolina at Chapel Hill, Chapel Hill, North Carolina, United States of America, 5 Department of Respiratory Medicine, KU Leuven, Leuven, Belgium, 6 Machine Learning Department, School of Computer Science, Carnegie Mellon University, Pittsburgh, Pennsylvania, United States of America

* zivbj@cs.cmu.edu

**Data Availability Statement:** The processed Nuclear-Seq data is available at https://zenodo.org/record/8040906, under the DOI 10.5281/zenodo.8040906. Two code repositories associated with this article can be found in https://github.com/

## Abstract

Age prediction based on single cell RNA-Sequencing data (scRNA-Seq) can provide information for patients' susceptibility to various diseases and conditions. In addition, such analysis can be used to identify aging related genes and pathways. To enable age prediction based on scRNA-Seq data, we developed PolyEN, a new regression model which learns continuous representation for expression over time. These representations are then used by PolyEN to integrate genes to predict an age. Existing and new lung aging data we profiled demonstrated PolyEN's improved performance over existing methods for age prediction. Our results identified lung epithelial cells as the most significant predictors for non-smokers while lung endothelial cells led to the best chronological age prediction results for smokers.

## Author summary

Aging is characterized by several changes at the cellular and molecular levels. The type and rate of these changes varies between individuals and does not always correspond to their chronological age. Determining the 'molecular age' of an individual can therefore provide critical information about their susceptibility to various diseases and enable better treatment via personalized medicine. With accumulated data profiling the gene expressions for human lungs at various ages, we have developed machine learning methods to learn the individual molecular age. As we show, the transcriptome in an individual lung allows our method to accurately predict chronological age of the donor. We identified different cell types that correlate well with aging and showed that these cells consistently display aging artifacts across individuals and datasets. Specifically, we found that lung epithelial cells provide the best aging predictions for non-smokers and endothelial/smooth muscle cells as the best aging predictor for smokers. Our approach further revealed important apoptotic genes involved in the aging process of lung tissue.

alexQiSong/lung_aging_analysis, and (https://github.com/alexQiSong/polyEN.

**Funding:** This work was partially supported by National Institutes of Health grant nos. OT2OD026682, 1U54AG075931 and 1U24CA268108 to Z.B.-J. and by the National Institute of Health grants nos. U01HL145567, R21HL161723, R01HL127349 to N.K. The funders had no role in study design, data collection and analysis, decision to publish, or preparation of the manuscript.

**Competing interests:** The authors have declared that no competing interests exist.

## Introduction

The ability to predict chronological age based on the gene expressions in different cells or tissues is important for understanding the complex nature of aging and for personalized medicine. The aging process has been studied for several decades and is characterized, molecularly, by accumulated errors at various levels of regulation. Age-related molecular damage accumulation is associated with induction of cellular senescence [1]. At the gene level, transcription error rate increases during cell senescence, leading to disease phenotypes [2]. Additionally, telomere shortening [3] and epigenetic regulation [4] are also well-known molecular mechanisms modulating the cellular senescence processes. Particularly, epigenetic regulation is evidenced by strong correlation between DNA methylation changes and longevity of a wide range of organisms [5]. At the cellular level, cell senescence is usually coupled with cell cycle arrest, characterized by stagnation in the G1 phase of the cell cycle [6,7]. Cell communication is also known to be involved in this process. For example, the senescence-associated secretory phenotype (SASP) proteins secreted by cells can bind to surface receptors of surrounding cells and promote senescence in those cells [8]. Various cytokines have also been suggested for playing a key role in promoting senescence through cell-cell interactions [9].

During aging, lung tissue undergoes substantial transcriptomic changes and phenotypic changes [10–12]. Previous studies have also reported associations between aging and lung diseases such as idiopathic pulmonary fibrosis (IPF) [13], chronic obstructive pulmonary disease (COPD) [14], and the recent worldwide outbreak of COVID-19 [15]. Individuals of higher chronological age are usually at higher risk of developing these diseases or suffering from a higher mortality rate. Determining the 'molecular age' of an individual can therefore provide critical information of their susceptibility to various diseases and conditions and improve personalized treatment. A number of regression-based aging "clocks" trained on DNA methylation or plasma protein levels have been described in the literature [16–18]. These clocks accurately predict chronological age but cannot determine organ or tissue specific conditions. Transcriptomics-based age signatures provide both a set of genes (and / or cell types) associated with the aging process and the ability to explicitly associate molecular age with an individual. Several methods were developed to predict individual age from transcriptomic signatures. Fleischer et al applied an ensemble LDA (linear discriminant analysis) model to human dermal fibroblast cells for age prediction [19]. RAPToR is another tool that reconstructs an aging model using Independent Component Analysis (ICA) or Principal Component Analysis (PCA) followed by a generalized additive model [20]. However, these methods have mainly been applied to bulk data and do not utilize the actual distribution of genes within a tissue / organ. Such distribution may be very important for determining the age given the changing fraction of senescent cells. Recently, Buckley et al. described aging clocks trained on single-cell transcriptomic data from murine brain tissue [21] and Yu et al presented an aging status classification model for binarized single cell data [22]. These studies either utilized simple linear regression models that do not account for nonlinear expression changes, or the model does not directly predict chronological ages for individuals [22].

To overcome these issues, we developed a novel framework, PolyEN (polynomial elastic net) for modeling and predicting chronological age from single cell transcriptomics aging data. Specifically, we model gene expression over time as a continuous curve and learn a distribution for each point on the curve. We next combine curves using an elastic net regression model to select features and predict the chronological age using cell type specific data. We used the Human Lung Cell Atlas (HLCA), a recent collection of large-scale lung single cell datasets that includes lung samples from more than 100 healthy and diseased donors [23] to test our method and compare it to existing methods. As we show, our method outperformed

all previous methods when tested with HLCA data. We have also performed additional large-scale validation by applying the method to a new Nuclear-Seq dataset we profiled from human lungs. The results show that a number of cell types and genes are consistently identified as significant across the different datasets, indicating that they can serve as accurate and robust lung aging signatures.

## Materials and methods

### Ethics statement

Human lungs were collected following local hospital (UZ Leuven) ethical committee approval (ML6385) and informed patient consent. According to Belgian legislation, declined donor lungs can be used for research purposes.

### Nuclear-seq data profiling

Explanted lungs from organ donors unsuitable for transplantation were collected from University Hospitals Leuven, as previously described [24]. Lungs were sliced in 2 cm thickness from apex to base in the transverse plane while kept frozen, with systematic sampling of cylindrical tissue cores (1·4 cm diameter). Three tissue cores per lung were randomly selected from the upper, middle, and lower regions of the lung. Samples were then divided into 5 slices which were used for downstream processing and analyses.

### Methylation data processing

Whole methylome data was performed on one slice of each sample. Samples were obtained from the same set of donors we used to generate Nuclear-Seq data. Bulk RNA-Seq Methylation was assessed using the Infinium Human Methylation 450K Bead Chip (Illumina, Inc). A total of 865918 CpGs were present in the dataset. At each CpG site, methylation was reported as a $\beta$ value, which is the proportion of signal obtained from the methylated beads over the sum of signals from all beads. $\beta$ values ranged from 0 (no methylation) to 1 (full methylation). Pre-processing was performed per the manufacturer's protocol. Data was normalized to internal controls. Signals corresponding to probes with a detection p-value >0.05 were excluded from further analysis.

### Data preprocessing

Nuclear-seq data was filtered to remove low quality nuclei, defined as having low UMI counts or having mitochondrial gene contamination, and multiplets at the subject level. Data was then integrated using Seurat and clustered by Louvain clustering. We retrieved HLCA dataset from cellxgene website: https://cellxgene.cziscience.com/collections/6f6d381a-7701-4781-935c-db10d30de293 [23]. The downloaded data object contains log-normalized expressions which were used for all analyses. We further used the following additional datasets: 1) Disease and control group for the IPF cell atlas dataset [25]; 2) Carraro basal cell dataset from Gene Expression Omnibus (GEO) with accession number GSE143706 [26]. Following the standard single cell data processing scanpy protocol (https://scanpy-tutorials.readthedocs.io/en/latest/pbmc3k.html), we used the raw counts from the control and disease group of IPF cell atlas, Carraro, and Nuclear-Seq datasets and performed log normalization with size factor = 10000.

### Cell type transfer and data integration

**Cell type transfer.** To ensure consistency in cell type assignments, we transferred the HLCA cell types to all other datasets using scArches pipeline available at https://github.com/

theislab/scarches [27]. scArches maps a query dataset to a reference dataset by adding new batch nodes to its neural network trained on the reference dataset and performing transfer learning that retrains the model on the query dataset. We used the default parameters and default reference model (which is the HLCA dataset itself) of the scArches pipeline for model training and cell type transfer. HLCA provides five levels (ann level) of cell types in which higher levels indicate more refined subpopulations of cells.

**Cell type annotation for Nuclear-Seq data.** We performed manual cell type annotation for Nuclear-seq data. Nuclear-seq UMI count matrix was clustered by Louvain clustering and markers for each cluster were determined. Cell types were annotated based on known canonical markers from previously published single cell data. While some canonical markers for cell types in single cell analysis do not translate to single nuclear analysis (e.g. AGER for AT1 cells), other markers exist to define these cell types (e.g. RTKN1 for AT1 cells).

**Data integration.** To integrate IPF cell atlas control data and Carraro control data with HLCA data, we performed data integration using the mnnpy pipeline available at https://github.com/chriscainx/mnnpy. mnnpy integrates different single cell datasets using mutual nearest neighbors [28]. The query dataset is then integrated into the reference dataset by multiplying with a correction vector (See S1 Text for more details).

## Model training and testing

**Feature extraction as polynomials.** Since age is a function of an individual and not a function of cell, any method for predicting aging needs to combine information from all cells (or at least all cells of the same type for cell-type-specific predictions). Compared to existing methods, our method utilizes the following steps to combine information from all cells. First, we included the expression variances since previous studies have shown that older individuals have more variable gene expression, even for the same cell types [29–31]. Second, we used polynomials to combine measurements across time. Such modeling allows for 'smoothing' of the individual time points (ages) which helps in reducing overfitting, especially when dealing with heterogenous data from several different individuals. Some genes exhibit expression changes more than other genes as part of the aging process [32]. To model aging over time, we assumed that genes involved in aging can be represented by 1st and 2nd degree polynomials. This assumption can identify genes that are monotonically increasing or decreasing with age as well as genes that participate in a subset of the aging process and then decrease in expression (we also tested the use of a 3rd degree polynomial but did not observe improvements). Specifically, we computed linear and quadratic polynomials for both the mean and the variance of each gene over all donors for each cell type and use these as inputs to a supervised regression model for age prediction. The resulted column-wise concatenated feature matrix is summarized below:

$$\{\boldsymbol{u}_1, \ldots \boldsymbol{u}_n, \boldsymbol{v}_1, \ldots \boldsymbol{v}_n, \boldsymbol{u}_1^{\circ 2}, \ldots \boldsymbol{u}_n^{\circ 2}, \boldsymbol{v}_1^{\circ 2}, \ldots \boldsymbol{v}_n^{\circ 2}\}$$

Where $\boldsymbol{u}_i, \boldsymbol{v}_i, \boldsymbol{u}_i^{\circ 2}, \boldsymbol{v}_i^{\circ 2}$ are column vectors that represent polynomials for ith gene. Specifically, $\boldsymbol{u}_i$ represents ith gene's donor-level mean expressions for a particular cell type and $\boldsymbol{v}_i$ represents ith gene's donor-level expression variance for a particular cell type. $\boldsymbol{u}_i^{\circ 2}$ and $\boldsymbol{v}_i^{\circ 2}$ represents second degree features (element-wise product) for $\boldsymbol{u}_i$ and $\boldsymbol{v}_i$ respectively. $\boldsymbol{n}$ is equal to number of input genes.

Expression means and variances were computed for each donor and each cell type separately. A complete input matrix for a particular cell type is a $n$ by $m$ matrix where $n$ represents number of donors and $m$ represents total number of polynomial features. We used a number of different lists from the literature to select a subset of the genes to fit as follows: 1) Fridman

senescence marker gene list [6]; 2) SASP2 senescence marker gene list [8]; 3) SenMayo senescence marker gene list [33]; 4) CellAge senescence marker list [34]; 5) The union of the above senescence markers; 6) AgeAnno differentially expressed genes for alveolar cell, B cell, endothelial lymphatic cell, epithelial cell, fibroblasts cell, mast cell, myeloid cell and T cell [35]; 7) All expressed genes in HLCA dataset. These marker genes can be found in S1 File. Since the number of genes in several of these lists is large (on the order of several hundreds or thousands) and training sample sizes are relatively small (10~20 donors), we further tested whether using PCA could improve performance. We transformed the original gene dimensions into PCA space which preserves the top 10 principal components (PCs), then feature computation described above was performed on the top 10 PCs (n = 10 in this case).

**Model training and hyperparameter tunning.** We used elastic net [36] to perform model training. The objective function is defined by the following equation:

$$\min_{w} \frac{1}{2n} \|Xw - y\|_2^2 + \alpha\rho\|w\|_1 + \frac{\alpha(1-\rho)}{2}\|w\|_2^2$$

Where $X \in \mathbb{R}^{n \times m}$ is the input polynomial feature matrix, $y \in \mathbb{R}^n$ is the donor age vector and $w \in \mathbb{R}^m$ is the vector of learned parameters. We tuned two hyperparameters $\alpha$ and $\rho$ on the entire training set. These parameters control the L1 regularization strength and the ratio between L1 and L2 regularization. Specifically, $\alpha$ was searched within the set {0.001, 0.01, 0.1, 1, 10, 100} and $\rho$ was searched within the continuous interval [0.1,1.0]. We used python package hyperopt to perform the optimal hyperparameter search. Hyperopt performs a sequential model-based optimization which sequentially approximates the optimal hyperparameter set by a surrogate model [37]. We set maximum number of evaluations as 30 for Hyperopt based tunning.

**Model testing.** We adopted two testing methods in this study: 1) leave-one-out (LOO) test, which means training on all donors but leave one donor out for testing and repeating this for all donors. 2) cross-dataset (CD) test, for which we trained a model on several datasets and tested the trained model on hold-out datasets. LOO test was performed for HLCA dataset and the Nuclear-Seq dataset. For CD test, we adopted different training and testing partitions for smoker donors and non-smoker donors. For smoker donors, we used donors from the study "Banovich_Kropski_2020" [38] in HLCA as training data and all other donors in HLCA as well as smoker donors from the control group of IPF and Carraro dataset for testing. For non-smoker donors, we used all non-smoker donors in HLCA dataset except for the study "Banovich_Kropski_2020" as training data, and all non-smoker donors in the study "Banovich_Kropski_2020" as well as all non-smoker donors from Carraro and IPF control data as test data. For convenience, we use the following shorthand terminology to refer to the combined datasets described above throughout this study: HLCA_sub1 (non-smoker training set), C1 (non-smoker testing set), HLCA_sub2 (smoker training set), and C2 (smoker testing set). To keep the genes consistent between the training and testing set for CD tests, we only used genes expressed in both sets (after preprocessing steps described in the section **Preprocessing**). We used $R^2$ score to evaluate the model performance in LOO and CD test (See S1 Text for more details).

**Comparison to other methods.** Data used by all methods were preprocessed in the same way as described in previous sections. For the baseline linear regression with L1 regularization, we tuned the regularization strength $\alpha$ using the same method and search space we used for PolyEN. For the Fleischer et al method [19], we used class_size = 20, subset_fold = 5, subset_min = 5 following the guide as introduced in https://github.com/jasongfleischer/Predicting-age-from-the-transcriptome-of-human-dermal-fibroblasts. For RAPToR, we used the default parameters and settings for the training function ge_im().

**Feature importance and feature p-values.** To rank the importance of the genes selected by the model, we used a feature attribution method, called shapley additive explanations (SHAP) [39]. The SHAP value represents how much impact a change in the input can have on the output compared to using a set of reference examples as inputs (See S1 Text for more details).

**Age prediction based on methylation data.** To calculate epigenetic aging clock, methylation data was uploaded to https://dnamage.clockfoundation.org. This provided an output comprised of the Horvath, Hannum, and GrimAge epigenetic clocks for each sample which were used for comparative analysis.

## GO analysis

We used GSEApy package [40] to perform Gene Set Enrichment Analysis (GSEA) to all input genes for each cell type. GSEA can identify significantly enriched functional terms given a list of ranked genes. In this study, we used aggregated SHAP scores as ranking metric for the genes.

## Results

### Overview of the analyses and datasets

We developed, tested, and validated methods for predicting chronological age from large single cell transcriptomics datasets. Fig 1 presents the overview of our computational prediction method and the study design. Our method relies on a continuous model for the means and variances of the gene expressions in different cell types over time. To learn such a model, we fit different polynomials to means and variances extracted from the gene expressions. Next, we used the polynomial features learned by this process to train a regression model for predicting chronological age. The genes we used are either sets of senescence marker genes or all expressed genes in each dataset. We used either the original gene dimensions or PCA-transformed dimensions (Materials and Methods). We repeated these steps for each cell type and tested each cell type using a leave-one-out (LOO) test or cross-dataset (CD) test (Materials and Methods). We also used a method for feature ranking to identify the most informative genes, [39] and assigned empirical p-values to genes using a permutation-based approach (Materials and Methods).

We applied our framework to several different single cell lung datasets. These included the largest lung cell dataset that is currently available (Human Lung Cell Atlas, HLCA) [23] HLCA contains over 500000 cells from over 100 donors. After preprocessing (see Materials and Methods), we obtained 143576 cells and 26975 genes from 39 smoker donors, and 179314 cells and 27115 genes from 39 non-smoker donors (see Table 1). Additionally, for CD test, we retrieved control group donors from the idiopathic pulmonary fibrosis (IPF) cell atlas study [25], and control group donors from a study of IPF lung basal cells [26], which we will further refer to as the Carraro dataset. After preprocessing, for the IPF control dataset, we obtained 22378 cells and 33931 genes from 8 smoker donors, and 62469 cells and 48231 genes from 23 non-smoker donors (Table 1). For the Carraro dataset, we obtained 12700 cells and 17300 genes from 4 smoker donors, and 4258 cells and 16365 genes from 2 non-smoker donors. To further test agreement of model predictions between datasets, we also partitioned the HLCA dataset by studies (see Materials and Methods). Lastly, we also performed the same workflow to a new dataset we profiled using Nuclear-Seq dataset, which contains 46765 cells and 32551 genes from 3 smoker donors, and 384680 cells and 39756 genes from 21 non-smoker donors after the preprocessing steps. See S1 Text for discussion on the age distribution similarities and differences between the different datasets.

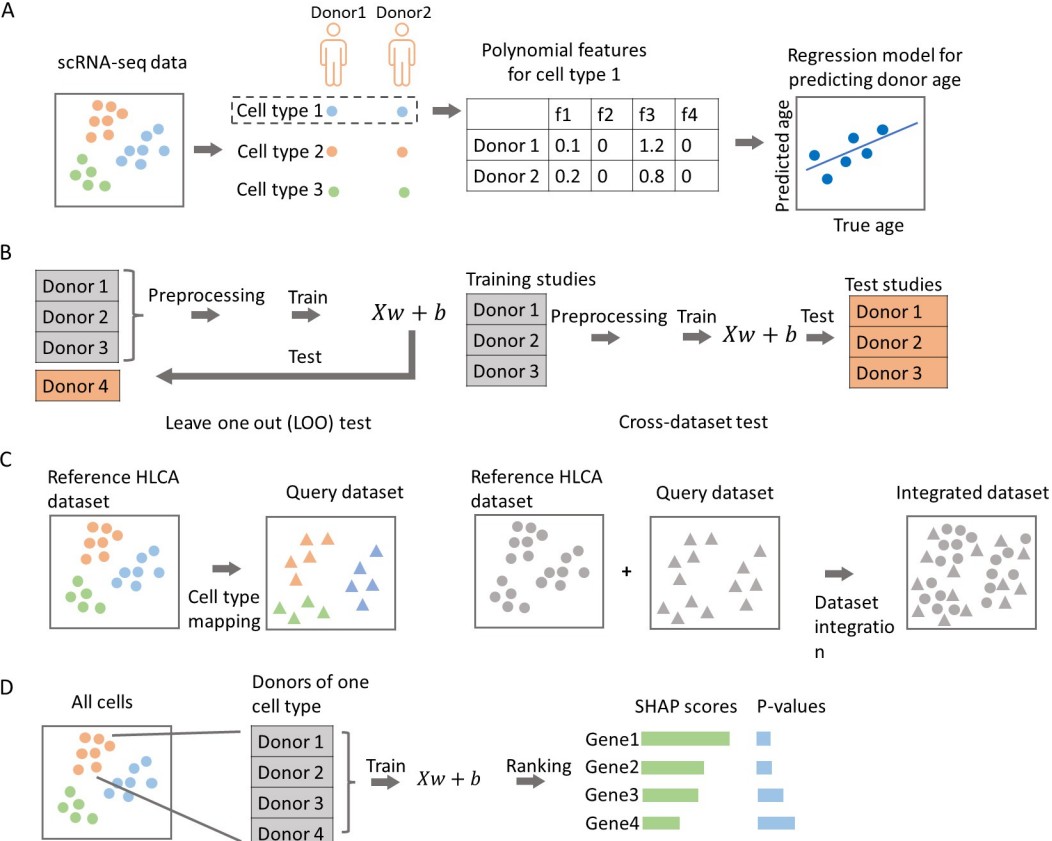

**Fig 1. The flowchart of the analysis. A**. Polynomial features were extracted for each cell type and features were aggregated at donor level. Regression models were trained based on the extracted polynomial features, either at original gene space or at the PCA-transformed space. **B**. Training and testing strategy, we adopted LOO test and CD test to evaluate the model performance (see Materials and Methods). **C**. Cell type mapping and dataset integration. **D**. SHAP-based ranking and empirical p-value for genes selected in each cell type.

## Cell type mapping and dataset integration

To keep the cell types consistent between training and testing dataset, we mapped cell types defined in HLCA to all other datasets using scArches pipeline [27] (see Materials and Methods). The hierarchical five-level cell types in HLCA can enable aging modeling at different granularity of cell populations, and identification of best marker genes at the corresponding resolution. The joint cell embeddings of Carraro dataset, the IPF control dataset, and HLCA show that in both smoker and non-smoker group, the cell type transfer was successful (Fig 2A, left) with uncertainty threshold of 0.2 (Table 2). However, the cell type transfer was not as

**Table 1. Number of cells, genes for each dataset after preprocessing steps.**

| Group | Type | HLCA | IPF | Carraro | Nuclear-Seq |
|---|---|---|---|---|---|
| Smoker | Genes | 26975 | 33931 | 17300 | 32551 |
| | Cells | 143576 | 22378 | 12700 | 46765 |
| | Donors | 39 | 8 | 4 | 3 |
| Non-smoker | Genes | 27115 | 48231 | 16365 | 39756 |
| | Cells | 179314 | 62469 | 4258 | 384680 |
| | Donors | 39 | 23 | 2 | 21 |

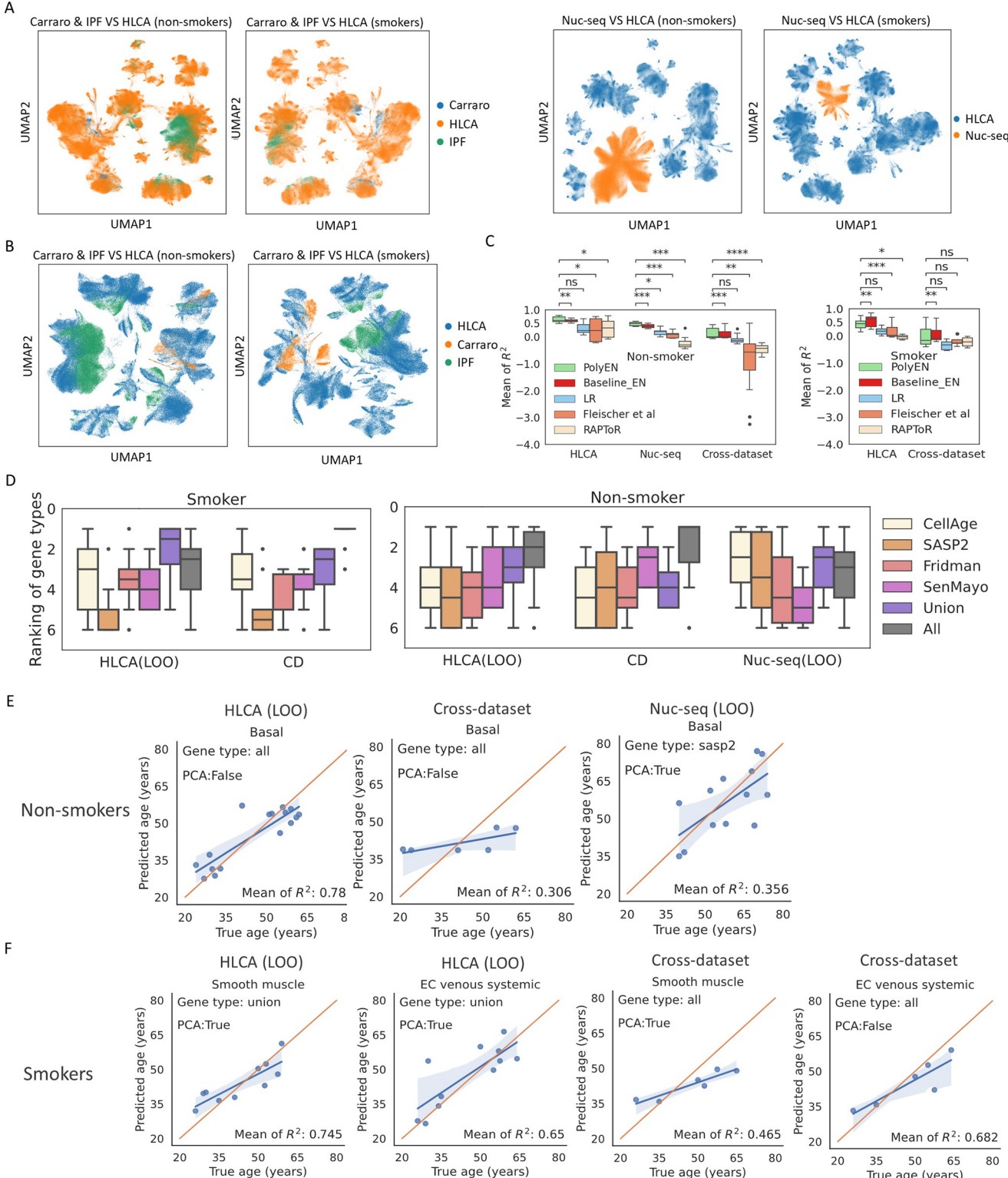

**Fig 2. Cell type mapping, dataset integration, comparison of gene types and cell types. A**. Joint cell embeddings of query datasets (IPF, Carraro, Nuclear-Seq) and reference dataset (HLCA) after cell type transfer performed by scArches. Plots were generated by the 30 dimensions of latent representations transformed from the original gene space. **B**. Joint cell embeddings of query datasets (IPF, Carraro) and reference dataset (HLCA) after dataset integration performed by mnnpy. Plots were generated based on the intersection of HVGs between the reference and the query datasets. **C**. The mean $R^2$ scores for the comparison of all tested methods. $R^2$ scores shown in the plot are from the top 10 cell types with highest $R^2$ scores for each method.

P-value annotation legend: ns (not significant): p-value $\geq$ 0.05; *: 0.01 < p-value $\leq$ 0.05; **: 0.001 < p-value $\leq$ 0.01; ***: 0.0001 < p-value $\leq$ 0.001; ****: p-value $\leq$ 0.00001. **D**. The rankings of different types of gene markers for the top cell types. The rankings were computed for each cell type separately. For each cell type, we selected for gene type's best PCA setting as determined by highest $R^2$ score. We used this $R^2$ score as the representative $R^2$ score of that gene type for the given cell type. We then ranked the different gene types by these representative $R^2$ scores. The resulted rankings (from 1 to 6) for the top 10 cell types are presented in the plots. These cell types are extracted from the top 10 cell types as shown in Figs 3 and. 4. **E,F**. Predicted donor ages VS true donor age for the top cell types. For each cell type, we selected its best gene type and PCA setting as determined by highest $R^2$ score. The corresponding best gene type and PCA setting is labeled in each plot.

successful for the Nuclear-Seq dataset (Discussion, Fig 2A, right and Table 2). Compared to IPF cell atlas control data and the Carraro dataset, we observed higher percentages of unannotated cells at all five levels in Nuclear-Seq dataset. We therefore resorted to manual cell type annotation for Nuclear-Seq dataset (see Materials and Methods).

To overcome batch effect between training and testing datasets, we performed unsupervised dataset integration in the original gene dimensions. We used HLCA as reference dataset and other datasets as query datasets. For this task we used the mnnpy package [28] and the intersection of highly variable genes (HVGs) between the reference and the query dataset as "anchors" to compute the correction vectors for integration. The integration results shown in Fig 1B indicate that the Carraro dataset and IPF control are well integrated with the HLCA dataset.

## Comparison of PolyEN to baseline and prior methods

We applied PolyEN to the HLCA data and have also compared the results to those obtained by baseline and prior methods. We used both LOO and CD cross-validation. Specifically, for each cell type, we compared our model to a baseline model for which only mean expressions of all expressed genes and linear regression (LR) model with L1 regularization are used. Additionally, we included another baseline, elastic net model, using the same inputs but without variances and second-degree features. We also compared to two previously published methods that utilized transcriptomics data, Fleischer et al [19], and RAPToR [20]. Results as presented in Fig 2C and Fig B in S1 Text show that PolyEN significantly outperformed all other methods for the nonsmoker group (Fig 2C, Mann Whitney U test). For example, p-value for comparison of PolyEN with LR across the datasets is $7.477 \times 10^{-4}$; When compared to the Fleischer et al method the p-value is $2.2 \times 10^{-24}$; and $4.99 \times 10^{-9}$ for the comparison to RAPToR. For the (much smaller) smoker group, PolyEN has again performed significantly better than all other methods except for baselineEN in the LOO test on HLCA dataset. For the CD test, PolyEN results are also better than the other methods except for baselineEN though the differences are not significant.

**Table 2. Percentage of unannotated cells for each HLCA-defined cell type level.**

| Group | Level | IPF + Carraro | Nuclear-Seq |
|---|---|---|---|
| Smoker | 1 | 0.13% | 16.08% |
| | 2 | 1.1% | 28.51% |
| | 3 | 18.73% | 41.82% |
| | 4 | 34.74% | 49.73% |
| | 5 | 38.65% | 49.95% |
| Non-smoker | 1 | 0.24% | 14.8% |
| | 2 | 1.39% | 24.87% |
| | 3 | 16.94% | 39.42% |
| | 4 | 33.86% | 45.85% |
| | 5 | 40.11% | 45.96% |

## Basal cells accurately predict donor age for non-smokers

Next, we explored the top cell types with highest $R^2$ scores from the PolyEN pipeline. For the LOO test in HLCA dataset, we obtained $R^2$ scores for 48 cell types from all five levels, of which 9 cell types achieved mean $R^2$ score greater than 0.5. The top 9 cell types include "Basal resting" (level 4, mean $R^2$ = 0.789, baseline $R^2$ = 0.392), "Basal" (level 3, mean $R^2$ = 0.783, baseline $R^2$ = -0.233), "Suprabasal" (level 4, mean $R^2$ = 0.762, baseline $R^2$ = 0.265), "EC venous systemic" (level 4, mean $R^2$ = 0.649, baseline $R^2$ = 0.154), "Club (non-nasal)" (level 5, mean $R^2$ = 0.606, baseline $R^2$ = 0.661), "Club" (level 4, mean $R^2$ = 0.595, baseline $R^2$ = 0.661), "Lymphatic EC mature" (level 3, mean $R^2$ = 0.564, baseline $R^2$ = -1.860), "Lymphatic EC" (level 2, mean $R^2$ = 0.556, baseline $R^2$ = -1.744), "EC general capillary" (level 4, mean $R^2$ = 0.544, baseline $R^2$ = 0.193). We noticed that level 4 cell types are more abundant in the top 9 cell types (5 out of 9) compared to other levels, suggesting refined cell types are generally more predictive of donor age than coarse cell types. We found the union of senescence and aging markers generally performed better than any single type of senescence / aging marker list in HLCA and Nuclear-Seq LOO test (Fig 2D). Interestingly, we found that "all expressed" genes is the best performing marker list for several top cell types for both HLCA and Nuclear-Seq LOO test and the HLCA CD test results in Fig 2D. This suggests that aging marker lists derived from the cellular senescence events may not fully capture the aging regulatory activities at organ or donor level. Additionally, our evaluation results based on LOO test indicate that AgeAnno marker list indeed achieved better $R^2$ scores than other marker lists in both HLCA and Nuclear-seq dataset, though the performance based on CD test was not better than other marker lists and AgeAnno markers are only available to a few cell types: alveolar cell, B cell, endothelial lymphatic cell, epithelial cell, fibroblasts cell, mast cell, myeloid cell and T cell (Fig J in S1 Text).

Our results indicate that a number of epithelial subtypes and endothelial subtypes are the most predictive cell types for non-smoker ages in HLCA dataset as mentioned above (Fig 3). Specifically, the three basal-related cell types are the top three ranked cell types. We next performed a CD test to validate whether basal cells are indeed predictive of donor aging state across different studies. Since basal cells are relatively less abundant among the donors in HLCA dataset (16 donors, Fig 3), we constructed a test dataset which combines 3 donors from HLCA dataset, 2 donors from the Carraro dataset, and 2 donors from IPF control dataset (see Materials and Methods). These donors all have sufficient basal cells for feature extraction and modeling and are all from independent studies not used for training. For this comparison we found "Basal" (mean $R^2$ = 0.306), "Club" (mean $R^2$ = 0.448), and "Club (non-nasal)" (mean $R^2$ = 0.449) cells as the top performing cell types. These cell types have higher $R^2$ scores than other cell types and they all produced mean $R^2$ scores greater than 0.3 (Figs 2D, 3, and Fig C in S1 Text). Finally, we performed LOO test on Nuclear-Seq dataset which again identified basal cells as one of the top cell types with mean $R^2$ score = 0.356, validating their ability for accurately predicting age (Figs 2D, 3, and Fig C in S1 Text).

Additionally, we also tested the performance of polyEN when all cell types are combined to train a meta regressor. We concatenated gene expressions of all cell types in a column-wise manner and performed LOO test following the same steps as we did for single cell types. Each gene / cell type combination was used as a potential feature. We performed PCA on the concatenated matrix and selected the first 20 principal components as inputs to polyEN. Even though the method had access to top genes across all cell types, we did not find an improved performance compared to PolyEN models trained on single cell types (Fig F in S1 Text) likely due to overfitting. The marker list with highest $R^2$ for Nuclear-Seq dataset under LOO test is SenMayo (mean of $R^2$ = 0.253 vs. all other marker lists for all available cell types combined in

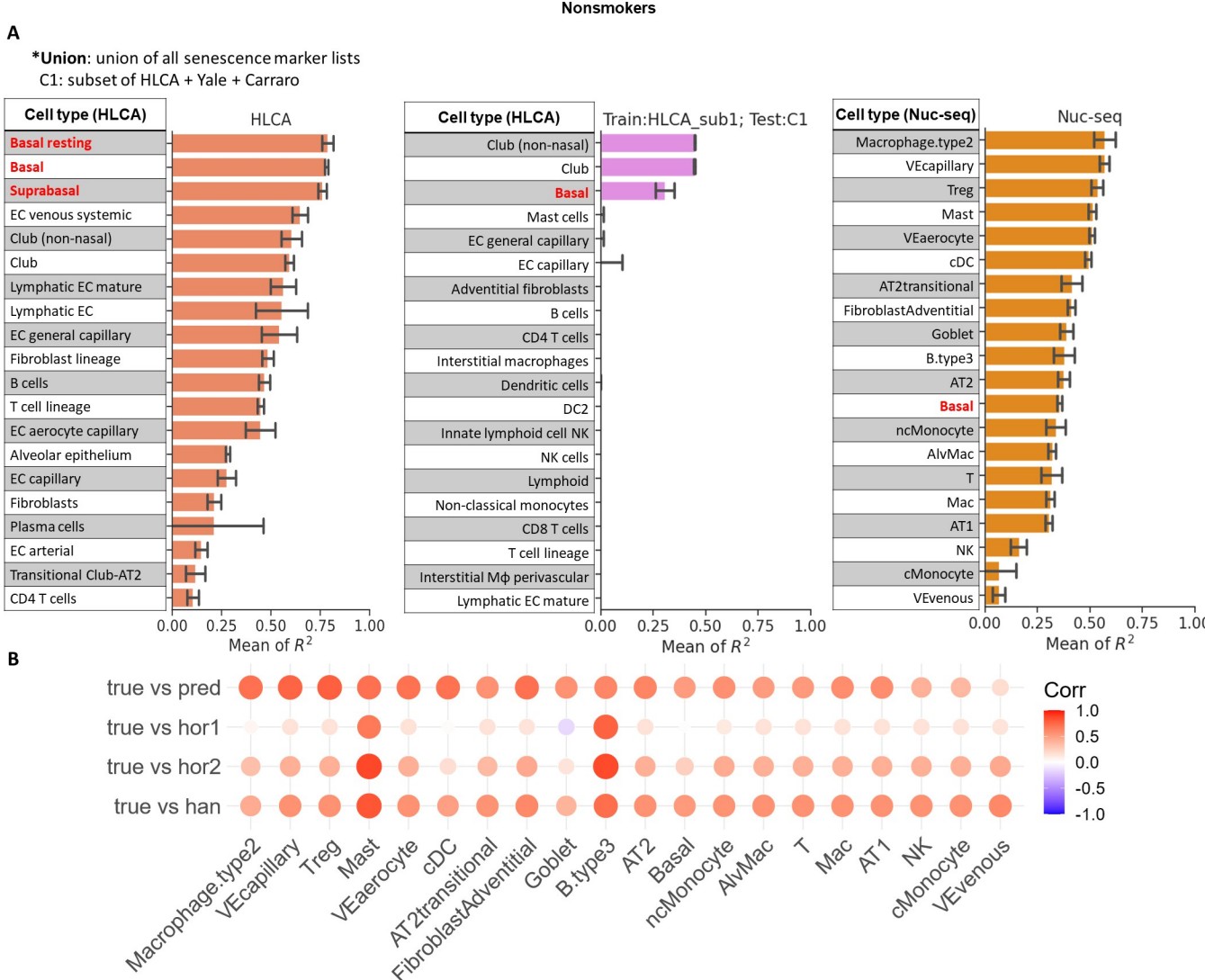

**Fig 3. $R^2$ scores from LOO test and CD test for non-smoker donors and comparison between transcriptome predicted ages and methylation predicted ages. A** $R^2$ scores for non-smoker donors tested with LOO test and CD test. Each row represents the corresponding best type of gene marker from a cell type. The bar in each row represents mean and standard deviation of $R^2$ score from five runs. Left: $R^2$ scores from the LOO test in HLCA dataset; middle: $R^2$ scores from the CD test which used a subset of HLCA for training and a subset of HLCA and other datasets for testing (Materials and Methods); right: LOO test in Nuclear-Seq dataset. The rows with no bars shown indicate $R^2$ scores equal to or smaller than zero. See S2 File for more information on the corresponding senescence markers and number of donors used in each row. **B** Comparison between transcriptomic ages and methylation ages. Transcriptomic ages were predicted by polyEN and methylation ages were predicted as described in Materials and Methods. "true" represents true chronological donor age and "pred" represents transcriptomic ages predicted by polyEN. "hor1" is methylation ages predicted by Horvath1 method; "hor2" is methylation ages predicted by Horvath2 method and "han" is methylation ages predicted by Hannum method. polyEN was applied to the cell types and marker gene lists shown in **A** for Nuclear-seq.

Nuclear-Seq data) and for HLCA the best marker list is all expressed genes (mean of $R^2$ = 0.118 vs all other marker lists for all available cell types at ann level 1).

## Smooth muscle cells and venous endothelial cells accurately predict donor age for smokers

We repeated the same procedure for donors of smoker group in all datasets except for the Nuclear-Seq dataset which does not include enough smoker donors (3 donors, see Table 1).

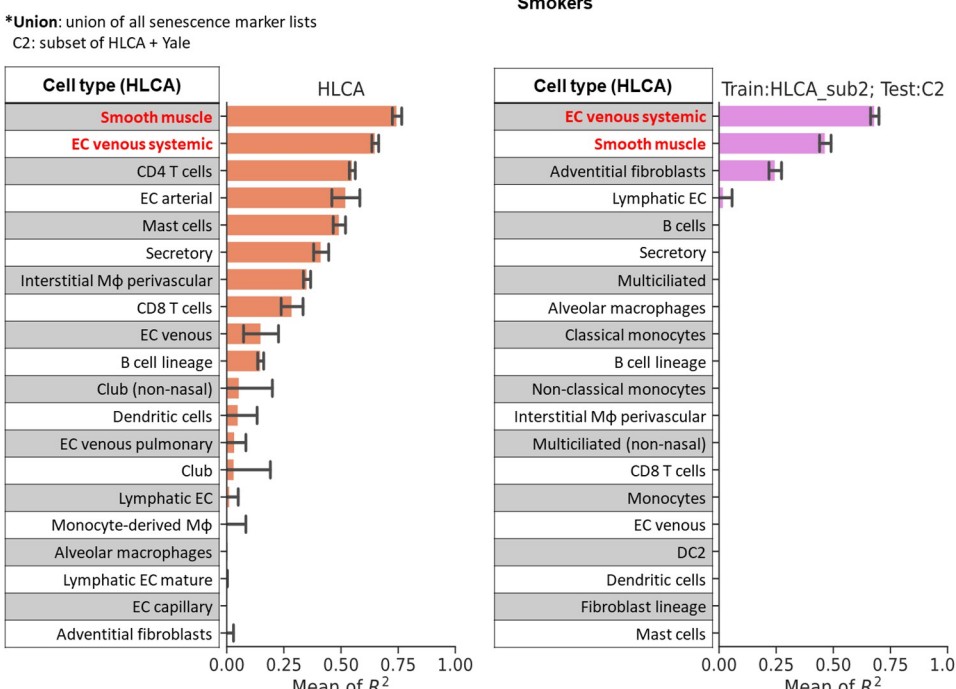

**Fig 4. $R^2$ scores from LOO test and CD test for smoker donors.** Each row represents the corresponding best type of gene marker from a cell type. The bar in each row represents mean and standard deviation of $R^2$ score from five runs. Left: $R^2$ scores from the LOO test in HLCA dataset; middle: $R^2$ scores from the CD test which used a subset of HLCA for training and a subset of HLCA and other datasets for testing (Materials and Methods). The rows with no bars shown indicate $R^2$ scores equal to or smaller than zero. See S2 File for more information on the corresponding senescence markers and number of donors used in each row.

For these donors we found there are four cell types that achieved $R^2$ scores greater than 0.5 in LOO test for HLCA dataset (Figs 4, 2F, and Fig E in S1 Text). These cell types include "Smooth muscle" (mean $R^2$ = 0.745, baseline $R^2$ = 0.223), "EC venous systemic" (mean $R^2$ = 0.650, baseline $R^2$ = 0.391), "CD4 T cells" (mean $R^2$ = 0.549, baseline $R^2$ = 0.151), "EC arterial" (mean $R^2$ = 0.521, baseline $R^2$ = -0.876), in which all but "EC venous systemic" achieved better $R^2$ with PolyEN models compared to all other methods. Prior work has shown that endothelial cells are involved in alveolar inflammation in COPD, a pulmonary disease usually caused by smoking [41]. The prominent role of endothelial cells in the smoker group during aging may suggest an increased risk of pulmonary disease. Next, we focused on constructing training and testing dataset to test smooth muscle cells in CD test. We included all donors from one study in HLCA which includes 5 donors with sufficient smooth muscle cells and all other donors in HLCA as test data. Cross datasets analysis of smooth muscle cells led to a high mean $R^2$ score ($R^2$ = 0.465 in Figs 4 and 2F). We also observed that EC venous systemic cells achieved results comparable to smooth muscle cells, with mean $R^2$ score equal to 0.65 from LOO test and mean of $R^2$ score equal to 0.682 from CD test (Figs 4, 2E, and Fig E in S1 Text).

Unlike the non-smokers, for the smokers, the best performing list was the union of senescence and aging markers (Fig 2D, LOO test in HLCA dataset). For the CD test, all expressed genes yielded slightly better results. Overall, the results in both non-smokers and smokers show that more genes can lead to better age predictions. Perhaps higher coverage of genes can capture the subtle changes of polynomials that marginally contribute to aging prediction.

## Smokers are predicted to be older when using non-smoker models

We tested the use of non-smoker models to predict chronological age for smokers based on their lung scRNA-Seq expression levels. We trained two different types of non-smoker models 1) using healthy non-smoker donors from the HLCA dataset and 2) using non-smoker donors with IPF disease from the IPF cell atlas (see Materials and Methods). We then used these models to predict the smoker chronological age in each respective dataset. The results showed that the model-predicted age distribution and the real age distribution are more consistent for healthy donors (Figs 5D and Fig H in S1 Text). However, for both datasets we observed an overall trend of greater predicted ages for smokers which may indicate that smoking increases the rate of lung aging in both healthy and diseased individuals (Fig H in S1 Text and [42]).

## Correlation between age prediction and methylation evidence

As mentioned in the Introduction, methylation was shown to be a strong predictor of aging [5]. We thus performed additional profiling and analysis of the Nuclear-seq data to calculate ages based on methylation data of each donor (See Materials and Methods). The Horvath, Hannum, and GrimAge epigenetic clocks were then compared to predictions based on our methods for various cell types identified in Nuclear-seq data (Fig 3A). While we observed positive correlations between predicted ages and real chronological ages for cell types shown in Fig 3A, transcriptomic ages predicted by polyEN are much more correlated with chronological ages when compared to methylation-based predictions, regardless of the epigenetic clock method being used (Fig 3B). This indicates that at least for lung cells, expression is a better age predictor than methylation. We also have explored pseudo-bulk expressions using PolyEN as an age predictor. For Nuclear-Seq data, we computed average expressions across all cells and performed training and testing following the same steps. Fig I in S1 Text showed that for the two top performing senescence marker lists "Union" and "CellAge", pseudo-bulk expression leads to a small improvement as compared to bulk methylation. This suggests that even without cell type information, transcriptome signatures may provide useful information about chronological ages, if marker gene list was chosen properly.

## Apoptotic genes revealed by importance scores

As mentioned above we used both senescence marker genes and all expressed genes as features for predicting chronological age. Senescence marker genes were obtained from previous publications [6,8,33,34] based on senescence cellular states and therefore may not fully capture the full regulatory programs related to aging at organ or donor level. As discussed above, we found that higher gene coverage usually led to better results in both smoker and non-smoker group (Fig 2D).

Interestingly, for the three top ranked basal-related cell types, the best gene type is consistently all expressed genes rather than any single type of senescence marker list. This may suggest that current senescence marker lists may miss some key genes related to aging in basal cells. To identify potential basal specific marker genes that might not be captured by senescence marker lists, we next performed gene set enrichment analysis (GSEA) for genes in these cell types. We first ranked genes by their SHAP scores (Methods) and assigned empirical p-values based on randomization tests. We then used the rankings as inputs to GSEA. Scores and p-values can be found in S3 and S4 Files and the full GSEA results for the basal related cell types can be found in S5 File. We found that genes selected in basal and basal resting cells are associated with cell cycle regulation, mitochondrial activities, mitotic regulation, and apoptotic signaling. These included categories such as GO:0001844 (p-value = 0.0, protein insertion into mitochondrial membrane involved in apoptotic signaling pathway), GO:0090266 (p-

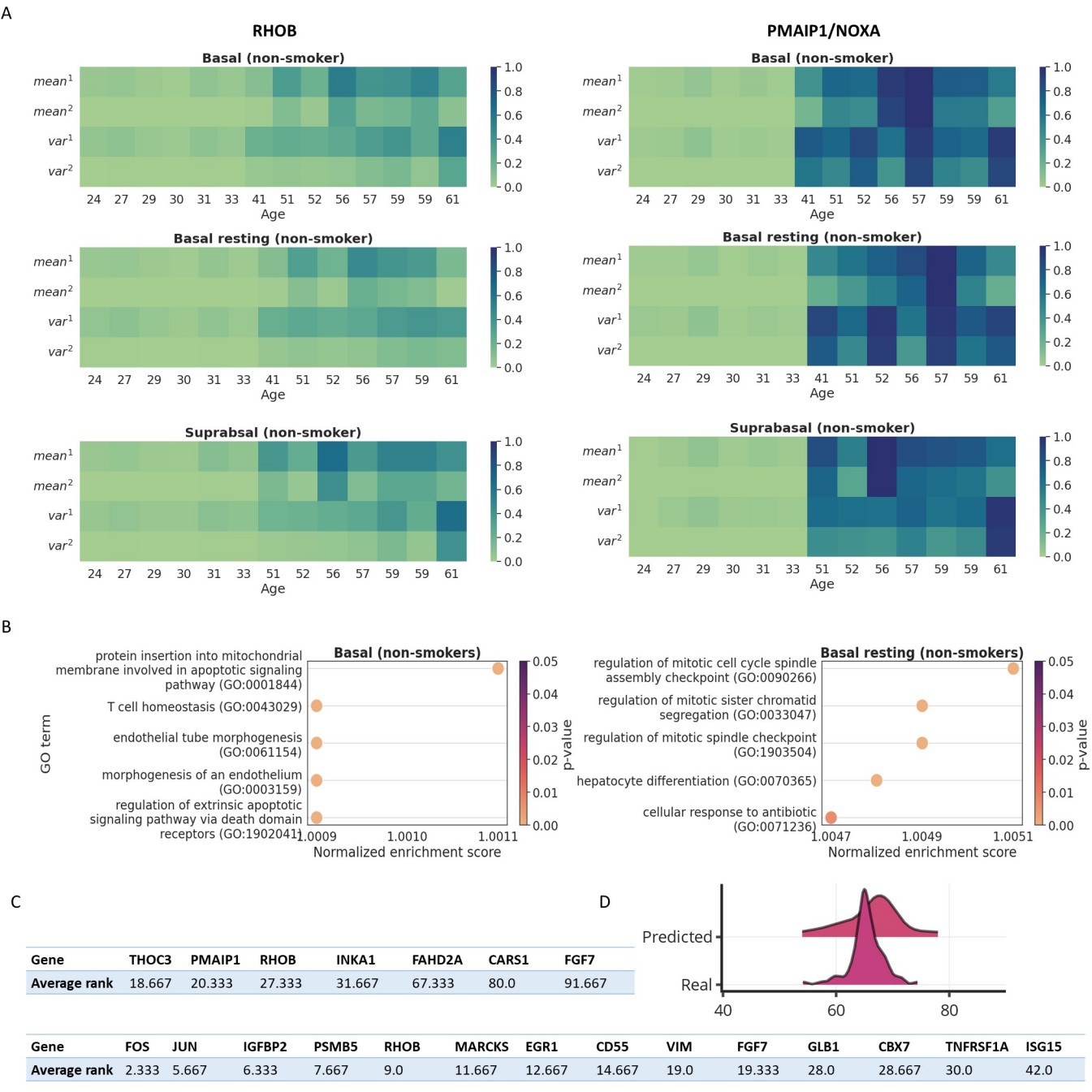

**Fig 5. Top significant GO terms from GSEA and polynomial features for basal-related cell types. A** Visualization of the polynomial features for *RHOB* and *PMAIP1*/Noxa in basal, basal resting and suprabasal cells of the nonsmoker group. Values visualized in the plots are polynomial features computed based on the log-normalized gene expressions. **B** The top five significant GO terms from GSEA for basal and basal resting cells of the nonsmoker group. **C** The common genes with significant SHAP scores among the three basal related cell types; top table: genes identified from all expressed genes; bottom table: genes identified from the union of senescence marker lists. **D** The distribution of predicted ages VS real ages for IPF disease donors. Models were trained using the non-smoker donors of IPF disease and tested using the smoker donors of IPF disease. x axis denotes the age and y axis denotes the density.

value = 0.0, regulation of mitotic cell cycle spindle assembly checkpoint), and GO:0033074 (p-value = 0.0, regulation of mitotic sister chromatid), (Fig 5A and S5 File). 8 genes were identified as leading-edge genes for the significant GO terms enriched in the three basal-related cell types (*PMAIP1*, *DUSP1*, *CD74*, *DNAJB1*, *ANXA1*, *CRIP1*, *KDF1*, *RHOB*). Of these, to date

only *RHOB* was listed as a senescence marker gene, suggesting that the top genes selected in the three basal-related cell types are not well captured by current senescence marker lists. Among these genes, *PMAIP1* appeared as a leading-edge gene in GSEA results for both basal and basal resting cells and was also selected as a significant gene for suprabasal cells. *PMAIP1* encodes Noxa, a known apoptosis activator which targets the mitochondrial Bcl2-regulated apoptosis pathway [43,44]. Overexpressed *PMAIP1* is preferentially localized to mitochondria. This is confirmed by our Nuclear-Seq dataset in which *PMAIP1* has a significantly lower score since Nuclear-Seq does not capture well mitochondria expression.

We also examined the differences between first-order versus second-order polynomial features. For each gene we computed the SHAP-based importance scores respectively for the four features (first-order, second-order expression means, and first-order, second order expression variances). We then used the top-ranked genes (Fig K in S1 Text) and calculated the weight of each type of feature in each gene. The results as visualized in Fig K in S1 Text show that for top-ranked genes, both types of features (first and second order) are needed. Many top-ranked genes show distinctive differences between the first-order and second-order features, suggesting that using higher-order features can capture additional knowledge about age related genes.

We also explored if SHAP-based ranking captured important genes by biasing towards highly expressed genes. To answer this question, we ranked genes by mean expressions in the three basal cell types shown in Fig 5. We observed very low overlap between the top-ranked gene sets from both methods. This shows that SHAP-based method has no strong bias towards highly expressed genes (Fig L in S1 Text).

We further compared the functions of gene ranking generated from SHAP-based method and gene ranking generated from model coefficients-based method. We performed GSEA analysis for the two gene rankings following the same steps that generate Fig 5B. Results for basal cell genes (Fig M in S1 Text) showed that SHAP-selected genes are enriched for more relevant categories compared to coefficient selected genes.

Additionally, we also explored the temporal expression patterns of the three basal-related cell types (See RESULTS section in S1 Text for more details)

## Discussions

We developed a new framework for age prediction from single cell RNA-Seq data. Unlike prior methods, our method utilizes both the average expression and the distribution information of gene expressions for age prediction. Application of the method to a large cohort of lung single cell data indicates that it can accurately predict the correct age and improves upon prior methods. Application to a new Nuclear-Seq dataset identifies novel cell types and genes predicted to be involved in the aging process.

Our results indicate that for lung, epithelial and endothelial cell types are more predictive of donor age than other cell types. This is consistent with previous studies that characterized senescent cell types in the lung [45–48]; We also observed a two-module pattern in basal related cell types. Previous studies reported three peaks of human plasma proteomics activities at ages of 34, 60, and 78 [32]. Our visualized polynomial features confirmed that the expression activities indeed go through substantial changes after age 34 (Figs 5A and 6). To explore this feature for individual genes, we further examined how the polynomial features of the *PMAIP1* and *RHOB* gene in basal related cells correlate to aging state. We selected the two genes because *PMAIP1* was identified as leading-edge gene by GSEA in both basal and basal resting cells, and *RHOB* is a common top gene as shown in Fig 5C. The temporal patterns of *PMAIP1* show a distinct two-module structure in which the polynomial features of *PMAIP1* drastically increased at and after age 41, suggesting elevated mitochondrial-targeting apoptotic

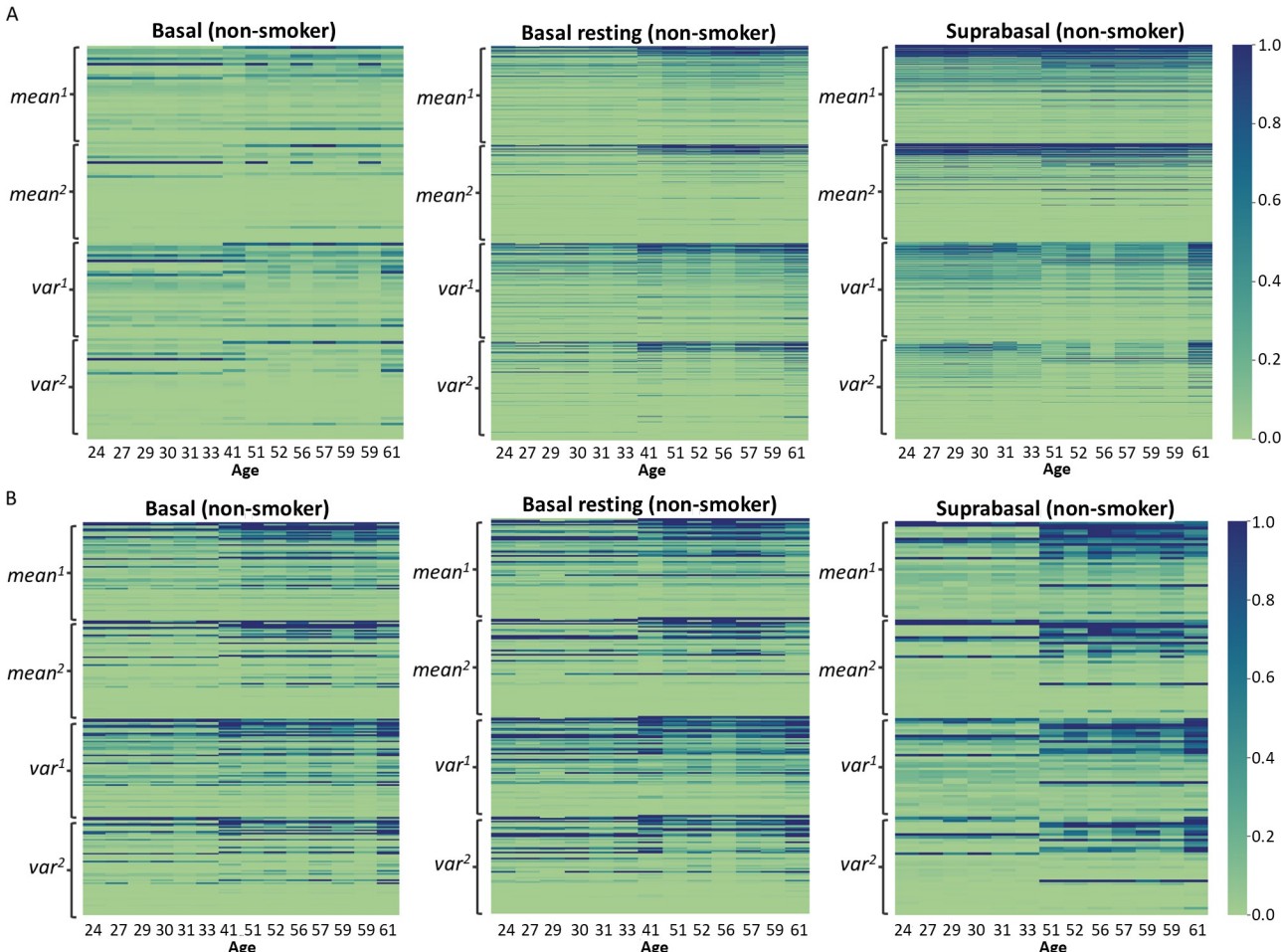

**Fig 6. Polynomial features for genes with significant SHAP scores identified in basal, basal resting and suprabasal cells of nonsmokers. A.** Visualization of the polynomial features for all expressed genes. We selected only the genes assigned with significant empirical p-values for each cell type (See Materials and Methods). **B.** Visualization of the polynomial features for union of senescence markers. We selected only the genes assigned with significant empirical p-values (See Materials and Methods). Each row represents one gene and genes were sorted by row-wise sum.

regulations (Fig 5A). However, since basal related cells are relatively less abundant in the HLCA dataset, the donor age coverage does not fully reveal the exact activation age of apoptotic signaling. Thus, we cannot rule out that up-regulation may occurred earlier than age 40, as previously reported [32]. We observed a similar pattern for *RHOB* (Fig 5A).

The observed difference between the Nuclear-seq dataset and the HLCA dataset in Fig 2A can be explained by the fact that we profiled biobank frozen samples using Nuclear-Seq data whereas most of the HLCA data was profiled from fresh tissues. The difference may also be due to different gene profiles present in the nucleus compared versus those in the cytoplasm. We further explored the transcriptomic difference between Nuclear-Seq data and the HLCA data. We identified the top-ranked marker genes of basal cells from the HLCA dataset and compared the expressions of these genes in Nuclear-Seq basal cells and HLCA basal cells. The results show there is a significant disparity between the two datasets (Fig N in S1 Text), suggesting that the cytoplasmic contents may have different molecular signatures compared to the nuclei for the same cell types.

While our method performs well, it still suffers from a few limitations. While PolyEN improved on all prior methods when predicting aging for the larger 'smokers' group, Elastic-Net performed better for the much smaller non-smoker set. This may imply challenges related to overfitting or to smoothing when dealing with a small and highly variable dataset (Fig H in S1 Text and [42]). Another issue is the use of individual cell types. A more complex model that captures aging from multiple cell types would be an important follow-up work. In addition, our model does not utilize any regulatory information and therefore may miss interactions that play an important role in aging.

In addition, we observed that cross-dataset validation did not achieve results comparable to leave-one-out cross validation. This is partly due to relatively small sample sizes for each cell type (See S2 File). When we further split the HLCA data into training and test sets, there are only limited donors available for each cell type in both training and testing sets, making cross-dataset evaluations very challenging. With more samples and further developments in methods, we may be able to see improved performance for cross-dataset benchmark.

Our analysis code is available at https://github.com/alexQiSong/lung_aging_analysis and we also provide a python script (https://github.com/alexQiSong/polyEN) to easily train and test polyEN model for any new single cell lung data. We hope that further refinement of these models will continue to shed light on this important prediction task.

## Supporting information

**S1 Text. The supplementary text and figures.**
(DOCX)

**S1 File. The detailed lists of senescence markers used in this study.**
(XLSX)

**S2 File. The detailed information for types of markers and number of donors used in Figs 3 and 4.**
(XLSX)

**S3 File. P-values of all significant genes for each cell type tested in non-smokers from the HLCA dataset.** Markers used were 'all expressed genes'.
(XLSX)

**S4 File. P-values of all significant genes for each cell type tested in smokers from the HLCA dataset.** Markers used were 'all expressed genes'.
(XLSX)

**S5 File. GSEA results for basal, basal resting, and suprabasal cells tested in non-smokers from the HLCA dataset.** Markers used were 'all expressed genes'.
(XLSX)

## Author Contributions

**Conceptualization:** Qi Song, Alex Singh, Ziv Bar-Joseph.

**Data curation:** John E. McDonough, Taylor S. Adams, Robin Vos, Ruben De Man, Greg Myers, Laurens J. Ceulemans, Bart M. Vanaudenaerde, Wim A. Wuyts, Xiting Yan, Jonas Schupp, James S. Hagood, Naftali Kaminski.

**Formal analysis:** Qi Song, Alex Singh, John E. McDonough.

**Funding acquisition:** Ziv Bar-Joseph.

**Methodology:** Qi Song, Alex Singh, Ziv Bar-Joseph.

**Project administration:** Ziv Bar-Joseph.

**Software:** Qi Song.

**Supervision:** Ziv Bar-Joseph.

**Writing – original draft:** Qi Song, John E. McDonough, Taylor S. Adams, Ruben De Man, Naftali Kaminski, Ziv Bar-Joseph.

**Writing – review & editing:** Qi Song, Alex Singh, John E. McDonough, Taylor S. Adams, Robin Vos, Ruben De Man, Greg Myers, Laurens J. Ceulemans, Bart M. Vanaudenaerde, Wim A. Wuyts, Xiting Yan, Jonas Schupp, James S. Hagood, Naftali Kaminski, Ziv Bar-Joseph.

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
