## [Decision Letter · Decision Letter 0]

15 Jul 2024

Dear Dr. Bar-Joseph,

Thank you very much for submitting your manuscript "Predicting Lung Aging using scRNA-Seq Data" for consideration at PLOS Computational Biology.

As with all papers reviewed by the journal, your manuscript was reviewed by members of the editorial board and by several independent reviewers. In light of the reviews (below this email), we would like to invite the resubmission of a significantly-revised version that takes into account the reviewers' comments.

We cannot make any decision about publication until we have seen the revised manuscript and your response to the reviewers' comments. Your revised manuscript is also likely to be sent to reviewers for further evaluation.

Sincerely,

Sushmita Roy, Ph.D.

Section Editor

PLOS Computational Biology

Sushmita Roy

Section Editor

PLOS Computational Biology

Reviewer's Responses to Questions

**Comments to the Authors:**

Reviewer #1: Authors developed a method named polyEN for prediction of ‘molecular age’ from single cell gene expression data. This method was benchmarked against existing methods in lung cell type data for smokers and non-smokers. The benchmarking is reasonable, and the results of the new methods are better than others. I have one comments about if expression is a better age predictor than methylation.

1. How to predict age for each cell type of lung from the methylation data is not clear. Did authors use single cell DNA methylation? A description of the methylation data used here is needed. Does the methylation data is profiled from the same cohort of individuals with the single cell RNA-seq data? I guess the three methods (Horvath, Hannum, and GrimAge) were trained on the bulk methylation data. If authors used single cell methylation data as input, how does it affect the results? The polyEN results are trained and tested on the similar datasets, however this is not the case for the methylation-based predictions.

Reviewer #2: Summary:

The paper presents an age prediction model for single-cell lung aging data sets. The method selects the list of gene sets from the literature related to senescence (or PCA-reduced dimensions of all genes) and calculates the mean and variance of the gene expression for the cells in a cell type. Next, it inputs first- and second-degree feature polynomials into an elastic net to predict the age of the donor's cell type as a regression task. The model is called PolyEN. After predictions, the paper performs SHAP analysis to learn relevant features for the model’s predictions. Some of the key insights from the work include: (1) PolyEN produces better results than existing bulk and single-cell age prediction models that perform regression (2) Basal cells accurately predict age for non-smoking individuals in the data (3) Smooth muscle cells and venous endothelial cells accurately predict donor age for smokers and (4) SHAP analysis reveals genes related to interesting gene ontology descriptions to understand aging better.

Major comments:

The idea of modeling the features as first- and second-degree polynomials is interesting, and the intuition behind it makes sense since the quadratic form could capture feature trends beyond the linear formulation. Investigating age regression performance that is consistent for certain cell types and gene sets helps understand aging in lungs among smokers and non-smokers. However, it would be helpful to include more detailed descriptions of the methodological setup and modeling choices to robustly support the results of the paper (see points below).

First, what was the intuition behind calculating means and variances instead of using single-cell gene expression read-out directly (as done in previous single-cell age prediction studies)?

For any normalization and dimensionality reduction done in the paper, was it ensured that it was done after the training/test split to avoid data leakage? Also, was a similar hyperparameter tuning scheme used to pick the best hyperparameters for the baseline models as well? What was the hyperparameter grid that was explored for them?

It is unclear why simple linear regression was chosen as a baseline when one could pick an elastic net with various ablations on the feature types as a baseline to better support the final formulation of the PolyEN model.

The scales in Figure 2C make it very hard to compare the boxplot performance. The cross-dataset results do not seem very good, raising questions about the model's generalizability.

It would be interesting to investigate important features obtained via the coefficients of the elastic net model. Do important genes overlap with the SHAP genes? I am also curious to learn how the paper is interpreting a SHAP importance score for a first-order versus second-order polynomial feature? Does it connect to the intuition behind using this formulation?

I could not read the text in Figure 3, which made it harder to understand the comparison of transcriptomics and methylation results.

Minor points:

A proof-reading pass of the paper (and the abstract) is highly recommended.

Higher-resolution figures with bigger font sizes are highly recommended.

CellBiAge was not included in the baseline comparison as it performed classification, but would the feature binarization step help the predictive performance on this dataset?

The main finding of the temporal analysis should be included in the main manuscript if it is mentioned. Also, its description in the Supplementary was difficult to follow.

**Have the authors made all data and (if applicable) computational code underlying the findings in their manuscript fully available?**

Reviewer #1: Yes

Reviewer #2: Yes

PLOS authors have the option to publish the peer review history of their article (what does this mean?). If published, this will include your full peer review and any attached files.

Reviewer #1: **Yes: **Zhana Duren

Reviewer #2: No
---

## [Decision Letter · Decision Letter 1]

23 Sep 2024

Dear Dr. Bar-Joseph,

Thank you very much for submitting your manuscript "Predicting Lung Aging using scRNA-Seq Data" for consideration at PLOS Computational Biology.

As with all papers reviewed by the journal, your manuscript was reviewed by members of the editorial board and by several independent reviewers. In light of the reviews (below this email), we would like to invite the resubmission of a significantly-revised version that takes into account the reviewers' comments.

Please note that there are still several concerns that need to be adequately addressed. 

We cannot make any decision about publication until we have seen the revised manuscript and your response to the reviewers' comments. Your revised manuscript is also likely to be sent to reviewers for further evaluation.

Sincerely,

Sushmita Roy, Ph.D.

Section Editor

PLOS Computational Biology

Sushmita Roy

Section Editor

PLOS Computational Biology

Reviewer's Responses to Questions

**Comments to the Authors:**

Reviewer #1: Authors addressed all my comments and I recommend to accept this manuscript.

Reviewer #2: Thank you to the authors for their detailed responses. Most of my points have been addressed. (1) I appreciate clarity on the decisions behind using mean and variance across donors instead of direct readout (2) The details of the preprocessing and hyperparameter tuning are also clear.

However, some of the points need more clarification.

(1) To follow up on reviewer 1’s question: The methylation clocks work with bulk methylation data. Given, that the polyEN method’s performance gives no improvement when cell type information is combined, it would suggest that methylation clocks (bulk data) and polyEN (single-cell data) may not be directly comparable. A more direct comparison would be if all the single-cell gene expression was averaged across all cell types to create bulk information and see if transcriptome is still a better signal than methylation. Even if the donors were the same, the bulk and single-cell data would have very different distributions.

(2) I had a really hard time understanding the response for comparing PolyEN with the elastic net model. It is unclear why the decision to present the ranking in Figure S7 was made instead of just presenting the R2 and RMSE value comparisons between the models. Could something like Figure 2(c) be plotted instead for this comparison?

(3) The additional analysis of the SHAP values with coefficients is interesting. However, the observation that the top-ranked genes are different does not address the question completely. Was there a GO term analysis done for these genes like the one done for SHAP genes? Do these genes report similar GO terms or processes as SHAP genes? If they are different, what would that suggest?

(4) Also, is it possible that SHAP analysis is picking on highly expressed genes (looking at Figure 5 and 6)? If one were to rank the genes based on mean expression levels (or variance) and compare them to those ranked based on SHAP values, would one find significant overlap?

(5) Finally, despite my recommendation to do a proofreading pass and authors’ response that this was done, I found several issues with the writing. See examples below:

“led to best results for smokers” in the last line reads abrupt and vague in the abstract.

The author summary” has grammatical errors - “data profiling the gene expression”, “levels of genes in a individual”, “individual lung …. their ages”, “and that … as the best…”

Line 52: “accumulated errors at various leves” reads a bit vague (what type of errors?).

“P” capitalized in line 88 for polyEN but small in abstract.

Line 157 “utilizes the following.” ends abruptly.

Line 223 “Fleischer et al” needs a citation?

“Overview of the analysis and datasets” and “Cell type mapping and dataset integration” should probably be in the Methods section and not Results.

Line 300: “P” caplitalized for p-value but small everywhere else.

Line 297: “three previously published methods”.. is followed by listing only 2 - Fleischer et al and RAPTOR

**Have the authors made all data and (if applicable) computational code underlying the findings in their manuscript fully available?**

Reviewer #1: Yes

Reviewer #2: Yes

PLOS authors have the option to publish the peer review history of their article (what does this mean?). If published, this will include your full peer review and any attached files.

Reviewer #1: **Yes: **Zhana Duren

Reviewer #2: No
---

## [Decision Letter · Decision Letter 2]

13 Nov 2024

Dear Dr. Bar-Joseph,

We are pleased to inform you that your manuscript 'Predicting Lung Aging using scRNA-Seq Data' has been provisionally accepted for publication in PLOS Computational Biology.

Best regards,

Sushmita Roy, Ph.D.

Section Editor

PLOS Computational Biology

Sushmita Roy

Section Editor

PLOS Computational Biology

Feilim Mac Gabhann

Editor-in-Chief

PLOS Computational Biology

Jason Papin

Editor-in-Chief

PLOS Computational Biology

Reviewer's Responses to Questions

**Comments to the Authors:**

Reviewer #2: Thank you for addressing all my concerns. I appreciate the new results and do not have any additional comments.

**Have the authors made all data and (if applicable) computational code underlying the findings in their manuscript fully available?**

Reviewer #2: None

PLOS authors have the option to publish the peer review history of their article (what does this mean?). If published, this will include your full peer review and any attached files.

Reviewer #2: No

---

## [Editor Report · Acceptance letter]

10 Dec 2024

PCOMPBIOL-D-24-00349R2 

 Predicting Lung Aging using scRNA-Seq Data 

Dear Dr Bar-Joseph,

I am pleased to inform you that your manuscript has been formally accepted for publication in PLOS Computational Biology. Your manuscript is now with our production department and you will be notified of the publication date in due course.

With kind regards,

Livia Horvath
